# Analysis of the Role of Water Saturation Degree in HTO, $^{36}$Cl, and $^{75}$Se Diffusion in Sedimentary Rock

**Miguel García-Gutiérrez \*, Manuel Mingarro, Jesús Morejón, Ursula Alonso**  **and Tiziana Missana**

CIEMAT, Physical Chemistry of Actinides and Fission Products Unit, 28040 Madrid, Spain
\* Correspondence: miguel.garcia@ciemat.es

**Abstract:** The aim of this study was to analyze HTO, $^{36}$Cl, and $^{75}$Se(IV) diffusion behavior in a sedimentary rock, which was obtained from the site initially selected for the emplacement of centralized temporal disposal of radioactive waste in Spain. Different experimental methodologies were used to analyze radionuclide diffusion in this rock: the through-diffusion (TD) method and the instantaneous planar source (IPS) methods. For the conservative tracers HTO and $^{36}$Cl, the effective diffusion coefficients, $D_e$, were obtained by the TD method, which was applied to the consolidated material taken at different orientations with respect to the bedding plane (parallel and normal). This revealed a negligible anisotropy of the rock. To analyze the effect of the water saturation degree, the IPS method was used, which was shown to be very suitable for evaluating these effects. For these tests, the samples were crushed, adequately hydrated, and compacted. The apparent diffusion coefficient, $D_a$, was determined for all the tracers at five different water saturation degrees. For each of the tracers investigated, the results indicated that, when the water saturation decreased from 100% to 60%, the $D_a$ also decreased by at least one order of magnitude: for HTO, this decrease was from $1.5 \times 10^{-9}$ to $2.3 \times 10^{-10}$ m$^2$/s; that for $^{36}$Cl was from $4.6 \times 10^{-10}$ to $2.8 \times 10^{-11}$ m$^2$/s; and that for $^{75}$Se was from $3.6 \times 10^{-11}$ to $8.3 \times 10^{-13}$ m$^2$/s. The experimental diffusion profiles of HTO and $^{36}$Cl could be satisfactorily fitted considering a unique diffusion coefficient, whereas the profiles of $^{75}$Se could not. This behavior is related to the existence of different species of selenium in the system, or to different retention mechanisms.

**Keywords:** radioactive waste disposal; radionuclides; transport; claystone; barrier materials



## 1. Introduction

Surface radioactive waste repositories are possible disposal areas where radionuclide migration could affect the environment because contaminants can reach the surrounding soils, even under conditions of a low water hydraulic gradient [1].

Unsaturated conditions are generally present in soils and surface materials, but they can also be expected under transient conditions in the clayey barriers of deep geological repositories of high-level radioactive waste (HLWR).

Compacted bentonite, which is the main geoengineered barrier in HLWR [2], is emplaced under unsaturated conditions. For at least 100 years after repository commissioning, the ventilation of drifts and shafts will be active, which will desiccate the walls of clay host rock and shot [3]. For instance, the formation of fractures associated with the desaturation of argillaceous media have been observed in underground research laboratories, such as Bure and Tournemire in France [3,4] or Mont Terri in Switzerland [5].

For the safety assessment of repositories and the development of the design of appropriate barriers to isolate contaminated soils from the water flow, it is necessary to understand the diffusion behavior of radioactive pollutants, to analyze the variation in their transport parameters under different conditions, and to provide quantitative data.

Several studies on radionuclide transport in compacted/consolidated clays under saturated conditions have been conducted [6–9]. Thorough reviews of the experimental

methodologies to determine diffusion coefficients and their analytical and numerical modeling are also available [10–13]. However, experimental studies of radionuclide transport in unsaturated rock are significantly scarcer. The reported values of diffusion coefficients are obtained by means of very different techniques, which makes it often difficult to compare the results from one system to another.

Using X-ray radiography in samples from Canadian shale, Nunn et al. [14] observed a reduction of about 20% in the effective diffusion coefficient values for iodide, with an average desaturation of the sample of 4%–7%. Diffusion of $^{134}$Cs through consolidated Callovo-Oxfordian claystone was studied at water saturation degrees ranging from 100% to 81% by osmosis suction (from 0 to 9 MPa) using a highly concentrated solution of polyethylene glycol. In these tests, the effective diffusion coefficient decreased by more than one order of magnitude from the fully saturated to the most dehydrated sample [15,16].

In porous media, where the main solute transport mechanism is diffusion, if the water content decreases, the diffusion coefficient is expected to decrease, too. However, the relationship between diffusion coefficients and water content depends, to a great extent, on the considered porous media [17]. Furthermore, if the decrease in water leads to a correspondent decrease in the diffusion coefficients, the measurement of very low diffusion values can be very time-consuming, and, thus, the application of reliable experimental methodologies is needed.

The objective of the present study was to analyze radionuclide diffusion in a natural fine-grained sedimentary rock (lutite) by considering the effects produced by the rock water saturation degree. This rock was obtained from the site initially selected for the Spanish centralized facility for the temporal storage of high-level radioactive waste (Villar de Cañas, Cuenca, Spain). No previous data related to radionuclide diffusion on this material were available.

Three different diffusants were considered: tritiated water (HTO), which is a conservative (i.e., non-sorbing) tracer, whose transport represents the transport of water; chloride; a conservative anion, which can suffer anion exclusion in clayey materials [18]; and selenium, which is in the form of selenite, an oxyanion, a slightly sorbing tracer. Selenium-79 is a key radionuclide for guaranteeing the safety of a high-level waste repository due to its toxicity, long half-life, and relatively high mobility.

The first diffusion tests were carried out with conservative tracers by means of classical through-diffusion (TD) tests [10,19] on the fully saturated consolidated rock. These tests aimed to obtain the effective diffusion coefficient, $D_e$, parallel and perpendicular to the bedding plane to analyze possible anisotropy.

To analyze in depth the effect of the saturation degree, the rock was crushed and mixed with enough water to reach five different water saturation degrees (100%, 90%, 80%, 70%, and 60%) and was compacted at the desired density. In this case, the instantaneous planar source (IPS) method was selected as a more appropriate methodology, as it was formerly applied to highly sorbing tracers, which usually show very low diffusion coefficients [1,7]. Furthermore, in this type of experiment, the initial conditions of the samples were maintained throughout the test to avoid moisture losses.

These studies help improve the experimental methodologies for determining the transport parameters of hazardous species in geological materials and contribute to the evaluation of the long-term safety of radioactive waste repositories.

## 2. Materials and Methods

The experiments were conducted under aerobic conditions.

### 2.1. Characterization of Rock and Sample Preparation

The samples selected for experimentation were obtained from an intact core of a borehole (9.90–10.30 m depth) drilled at the Villar de Cañas site. Before performing diffusion tests, the samples were analyzed using X-ray diffraction (XRD) (X'Pert MPD, Philips, Amsterdam, The Netherland) and scanning electron microscopy combined with energy-

dispersive X-ray spectroscopy (SEM/EDX) (EVO LS15/LZ-5, Zeiss, Jena, Germany). The semi-quantitative analysis performed by XRD showed the following mineralogical composition: 37% gypsum, 25% dolomite, 28% phyllosilicates, 8% quartz, and 2% accessory minerals. The average porosity (by Hg intrusion porosimetry) and the bulk density of the intact bore core were 29.79% and 2.2 g/cm$^3$, respectively. The average porosity of these samples varied between 3 and 35% depending on the mineral distribution, showing a high heterogeneity degree.

The phyllosilicate fraction was mainly composed of palygorskite, which is most probably the main sorbing mineral of the rock. The cation exchange capacity, CEC, of the rock fraction (<2 µm) was 24 ± 2 meq/100 g, and the N$_2$-BET surface area was 45 ± 1 m$^2$/g.

Figure 1 shows the SEM pictures of the main minerals identified in the rock: gypsum crystals (Figure 1a) and a dolomite plus palygorskite (fibrous material with a high sorption capacity) material (Figure 1b). More details on the mineralogy and chemistry can be found elsewhere [20].

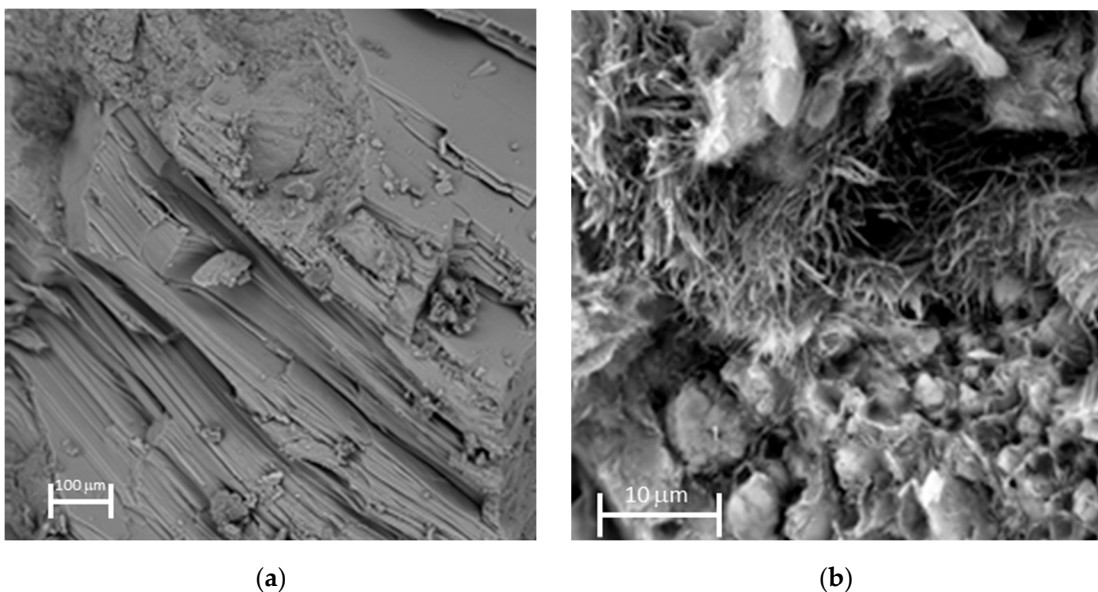

(**a**) (**b**)

**Figure 1.** SEM pictures: (**a**) gypsum crystals; (**b**) dolomite crystals and palygorskite (the fibrous material).

### 2.2. Equilibrium Water

Water in chemical equilibrium with the solid sample, to be used in all the tests, was obtained by combining 20 g of the crushed and sieved rock (φ < 1 mm) with one liter of deionized water and maintaining the suspension under constant stirring until reaching constant values of pH and electrical conductivity, σ (pH = 8.13 and σ = 1988 µS/cm). Table 1 shows the main chemical composition of this equilibrium water. The main ions present in this water were sulfate and calcium. Their concentration was related to the equilibrium with the gypsum present in the rock.

### 2.3. Tracers

Radioactive tracers were used in all the tests. Tritiated water, HTO, was supplied by PerkinElmer Spain, and $^{36}$Cl was supplied by Isotope Products Laboratories, Valencia, CA, USA. HTO and $^{36}$Cl are beta emitters. Their activity was measured by liquid scintillation counting with a Tri-Carb 4910TR PerkinElmer Liquid Scintillation Counter (LSC) using Ultima Gold as the scintillation cocktail.

A gamma emitter $^{75}$Se(IV) was supplied by Eckert & Ziegler (Valencia, CA, USA). It was measured using a Packard auto-gamma Cobra II 5003 counter with a 3″ NaI (Tl activated) crystal. Although the repository relevant isotope is $^{79}$Se, $^{75}$Se was used as it is the radioactive isotope that is commercially available.

**Table 1.** Chemical composition of equilibrium water (mol/L).

| Element | mol/L |
|---|---|
| $Cl^-$ | $1.52 \times 10^{-5}$ |
| $HCO_3{}^-$ | $1.51 \times 10^{-4}$ |
| $SO_4{}^{2-}$ | $1.35 \times 10^{-2}$ |
| $Ca^{2+}$ | $1.30 \times 10^{-2}$ |
| $Mg^{2+}$ | $2.60 \times 10^{-4}$ |
| $Na^+$ | $8.70 \times 10^{-5}$ |
| $K^+$ | $4.10 \times 10^{-5}$ |
| $SiO_2$ | $5.33 \times 10^{-5}$ |
| pH | 8.13 |
| Cond. (µS/cm) | 1988 |

*2.4. Experimental Set-Up of Diffusion Tests*

2.4.1. Through-Diffusion Method (TD)

Consolidated samples for through-diffusion (TD) tests were obtained by drilling a piece from the bore core with a diameter of 5 cm (both parallel and perpendicular to the bedding plane). The obtained cylinder was cut in slices of about 10 mm in thickness with a diamond saw.

TD tests were performed using the classical through-diffusion method as described elsewhere [6,21]. They were carried out with HTO ($7.0 \times 10^{-11}$ M) and $^{36}$Cl ($1.6 \times 10^{-6}$ M). The experiment lasted approximately 30 days. All tests were performed in duplicate.

For these tests, the consolidated sample, mounted within a stainless-steel diffusion cell (Figure 2), was located between two reservoirs (in-reservoir and out-reservoir), where the synthetic water was continuously stirred. After the time needed for the resaturation of the sample (approximately four weeks), the tracer was added to the in-reservoir. The concentrations in both reservoirs were kept constant, such that steady-state diffusion across the sample was achieved. To maintain the concentration gradient constant, a large (1 L) in-reservoir and a small (20 mL) out-reservoir were used and periodically changed.

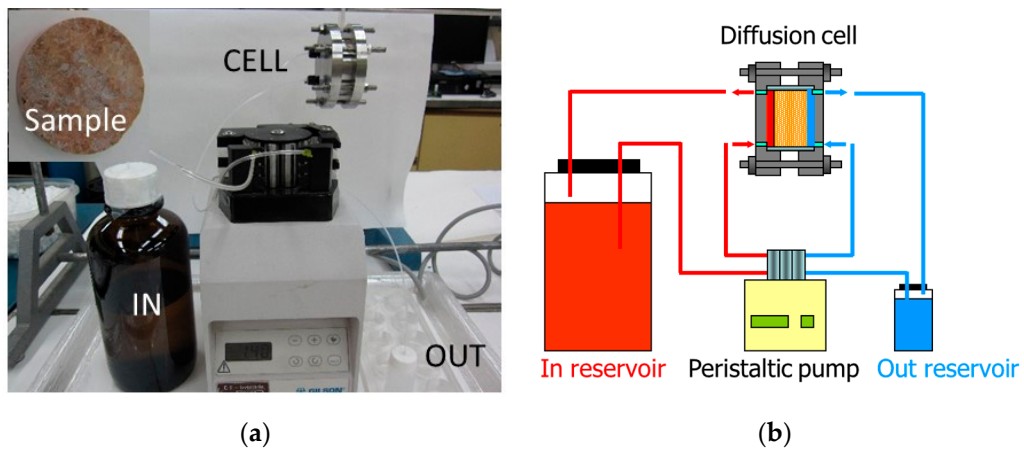

(**a**)          (**b**)

**Figure 2.** Experimental set-up for through-diffusion experiments: (**a**) Picture of the cell and reservoir. The image of one sample is included in the left corner. (**b**) Schematic design of the experiment.

To quantify the effective diffusion coefficient, an analytical expression was considered. For a sample with a thickness $L$, which is initially tracer-free and where the in-reservoir has a constant concentration, $C_0$, and the out-reservoir is kept to a "*quasi*" zero concentration, the expression of the cumulative mass of the tracer, $Q$, passed to the out-reservoir through a cross-sectional area, $A$, as function of the time, $t$, is [19,22]:

$$Q = A \cdot L \cdot C_0 \left[ \frac{D_e}{L^2} t - \frac{\alpha}{6} - \frac{2\alpha}{\pi^2} \sum_{1}^{\infty} \frac{(-1)^n}{n^2} exp\left( \frac{-D_e n^2 \pi^2 t}{L^2 \alpha} \right) \right] \tag{1}$$

After long time periods, a steady-state condition is reached. The series expansion in Equation (1) vanishes because the exponential term trends toward zero and a linear relationship is obtained between $Q$ and $t$:

$$Q = A{\cdot}L{\cdot}C_0 \left[ \frac{D_e}{L^2}t - \frac{\alpha}{6} \right] \tag{2}$$

where $\alpha$ is the rock capacity factor, defined as $\alpha = \varepsilon + \rho_d{\cdot}K_d$; $\varepsilon$ is the accessible porosity; $\rho_d$ is the dry density; and $K_d$ is the distribution coefficient. For non-sorbing tracers, $\alpha$ is equivalent to the accessible porosity. The effective diffusion coefficients, $D_e$, were calculated from the slope of the straight line, fitting the long-term behavior of $Q$.

### 2.4.2. Instantaneous Planar Source Method (IPS)

For studying the effects of water saturation on diffusion by the instantaneous planar source (IPS) method, the rock was crushed, sieved ($\phi < 1$ mm, American Society for Testing and Materials (ASTM) nº 18), and mixed with the equilibrium water necessary to obtain the five different saturation degrees (S): 60%, 70%, 80%, 90%, and 100%. The hydrated material was compacted into stainless-steel rings to the desired dry density (1.2, 1.4, and 1.65 g/cm$^3$).

Experiments with HTO ($1.1 \times 10^{-7}$ M) and $^{36}$Cl ($7.5 \times 10^{-4}$ M) were performed at a dry density of 1.65 g/cm$^3$. The time needed for these experiments was 6 h. In the case of $^{75}$Se ([Se] = $2.5 \times 10^{-4}$ M), IPS experiments lasted 4, 7, and 10 days for the dry densities of 1.2, 1.4, and 1.65 g/cm$^3$, respectively. All the experiments were performed in duplicate.

For the IPS method, a filter paper (Whatman nº 54) spiked with 0.03 mL of tracer solution (with this volume, the tracer is homogeneously distributed in the filter) was located between two identical compacted samples. Figure 3 shows the pictures of the cell assembled (Figure 3a) and disassembled (Figure 3b). The filter paper stuck to one side of the sample can be seen in Figure 3b. In this experimental configuration, the tracer diffuses from the filter through the pore water into the sample located on both sides of the filter along the axial direction.

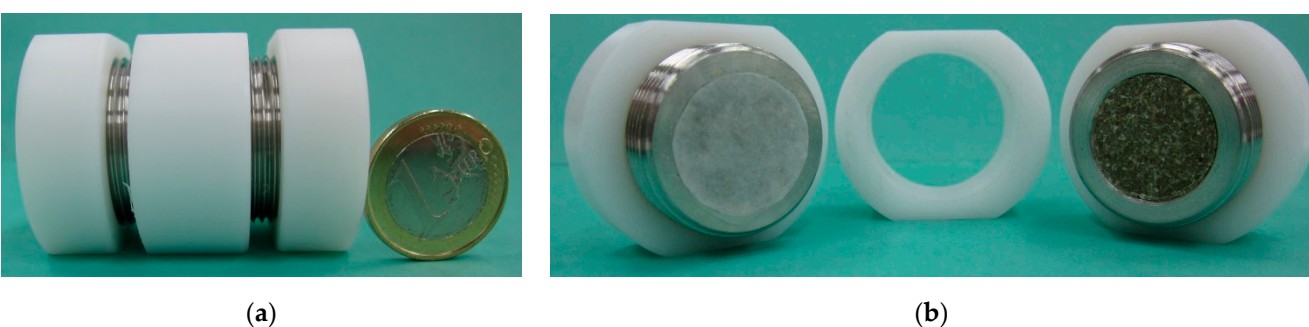

(**a**)                                                                                    (**b**)

**Figure 3.** Diffusion cell: (**a**) assembled cell; (**b**) disassembled cell.

After the diffusion time, the cells were dismantled and the sample was sliced in thin sections. The diffusion profile was obtained by measuring the radionuclide activity in each slice and knowing the position from the weight of each slice.

In the experiments with $^{75}$Se, the activity was measured directly in the solid using a Packard auto-gamma counter. HTO and $^{36}$Cl are beta emitters, and therefore their activity cannot be directly measured in the slices, but they must be extracted from the solid. The slice of the rock was suspended in 8 mL of deionized water and stirred for three days. After this time, the suspension was centrifuged ($25,000\times g$, 30 min), and $3 \times 2$ mL of the supernatant was taken for liquid scintillation counting.

The analysis of the experimental diffusion profile obtained with the IPS method was carried out while considering that the filter is an infinitesimally small thin source where

the tracer is homogeneously dispersed. The Dirac delta function, $\delta(x)$, can be used for the description of the tracer source at the initial conditions:

$$C(x = 0, t = 0) = M/(\delta(x) \cdot A) \tag{3}$$

$$C(-\infty < x < \infty, t = 0) = 0 \tag{4}$$

For this experimental configuration, an analytical solution [22] can be used to obtain the apparent diffusion coefficient, $D_a$:

$$C(x,t) = \frac{M}{2 \cdot A \sqrt{\pi \cdot D_a \cdot t}} exp\left(-\frac{x^2}{4 \cdot D_a \cdot t}\right) \tag{5}$$

where $C(x,t)$ is the radionuclide activity concentration (Bq·m$^{-3}$), $M$ is the absolute activity in the filter (Bq·m$^{-2}$) of the cross-sectional area $A$ (m), $x$ is the distance of each slice from the filter paper (m), and $t$ (s) is the experimental diffusion time. The values of $D_a$ (m$^2$/s) were obtained by fitting the experimental concentration profiles expressed as $C/M$ as a function of $x$ using least-squares adjustments.

## 3. Results and Discussion

### 3.1. Through-Diffusion Test

Figures 4 and 5 show examples of the results of the TD experiments obtained for HTO and $^{36}$Cl in the consolidated samples. The tracer's flux through the sample with the associated uncertainty and the cumulative activity, both versus time, was plotted. The best fit of the flux is shown as a continuous line. The linear fit of the cumulative activity was performed in the range of the steady-state regime (where the flux was constant).

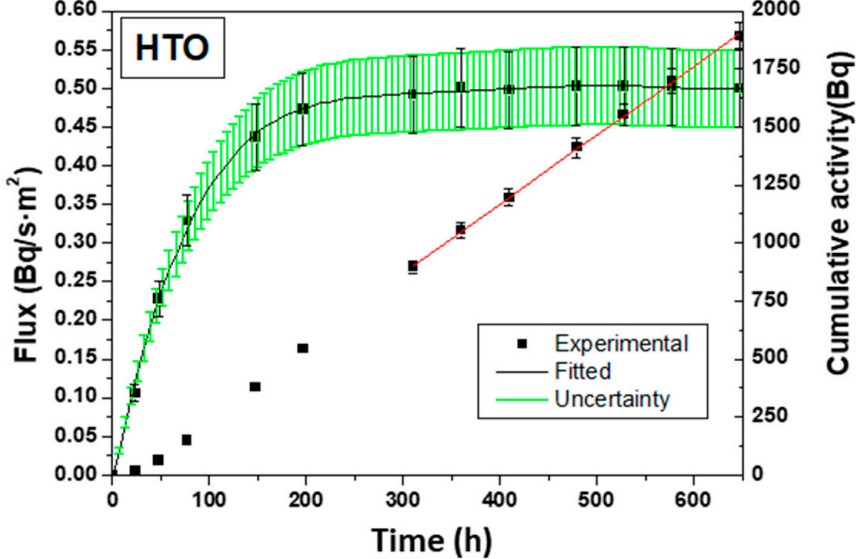

**Figure 4.** HTO: TD experiment perpendicular to the bedding plane of the consolidated sample. Flux and cumulative activity vs. time and linear fit of the steady-state region.

Table 2 shows a summary of all the effective diffusion coefficients, $D_e$, obtained by the fit of the experimental curves (paragraph 2.4.1) for the tracer diffusion parallel and perpendicular to the bedding plane. In addition, Table 2 shows the anisotropy factor ($D_{e(parallel)}/D_{e(perpendicular)}$) and the rock capacity factor $\alpha$. The uncertainties of the diffusion coefficients represent the standard deviation obtained after running duplicate experiments: the error of the anisotropy factor was calculated using the error propagation law.

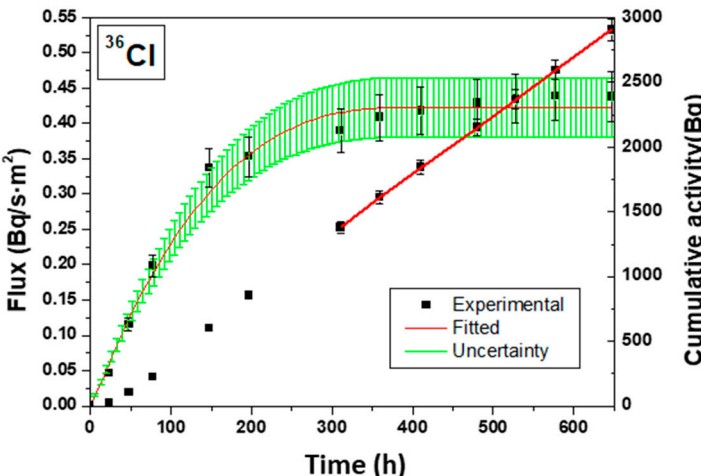

**Figure 5.** $^{36}$Cl TD experiment parallel to the bedding plane of the consolidated sample. Flux and cumulative activity vs. time and linear fit of the steady-state region.

**Table 2.** Effective diffusion coefficients, $D_e$ (m$^2$/s), of HTO and $^{36}$Cl on consolidated samples parallel and perpendicular to the bedding plane.

|  | **HTO** | **$^{36}$Cl** |
| --- | --- | --- |
| $D_e$ (m$^2$/s) Parallel | $(1.1 \pm 0.2) \times 10^{-10}$ | $(6.3 \pm 0.1) \times 10^{-11}$ |
| $D_e$ (m$^2$/s) Perpendicular | $(9.3 \pm 0.1) \times 10^{-11}$ | $(5.1 \pm 0.1) \times 10^{-11}$ |
| Anisotropy factor | $1.18 \pm 0.22$ | $1.23 \pm 0.14$ |
| $\alpha$ | $0.17 \pm 0.01$ | $0.07 \pm 0.01$ |

The effective diffusion coefficient values for HTO are half an order of magnitude higher than those of $^{36}$Cl. This may be related to the possible anion exclusion affecting chloride diffusion. Negligible differences were observed in the $D_e$ obtained in the samples parallel or perpendicular to the bedding plane for both HTO and $^{36}$Cl. For conservative tracers, $\alpha$ should be equal to the accessible porosity, $\varepsilon$. From diffusion experiments, the $\varepsilon$ obtained was 17% for HTO and 7% for $^{36}$Cl, which is affected by anion exclusion. The value of porosity obtained using diffusion tests was within the range of porosity measured by Hg intrusion.

Experiments formerly performed on consolidated Opalinus Clay from Mont Terri (Switzerland) by Van Loon et al. (2003) [6] showed a similar behavior and relationship between the diffusion of HTO and $^{36}$Cl, but with slightly lower $D_e$ values ($D_{e(HTO)parallel} = 5.4 \times 10^{-11}$ m$^2$/s; $D_{e(Cl)parallel} = 1.4 \times 10^{-11}$ m$^2$/s). They also found higher anisotropy, typical of the clay formation at Mont Terri. Similar results can be also found in [23] for different formation candidates being analyzed to host a deep geological repository.

### 3.2. Instantaneous Planar Source Tests with HTO and $^{36}$Cl

Figure 6 shows the diffusion profiles (normalized concentration vs. diffusion distance) for HTO and $^{36}$Cl at three different saturation degrees. The concentration profile for HTO, at any saturation degree, is wider than that obtained with $^{36}$Cl under the same conditions, in agreement with the previous tests on consolidated samples. The lower diffusion coefficient can be attributed to $^{36}$Cl anion exclusion. All diffusion profiles were successfully simulated using Equation (5).

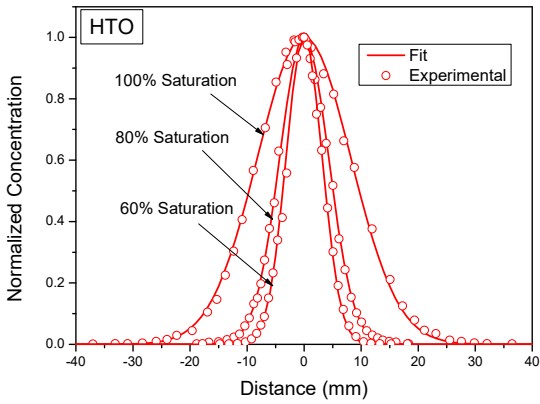

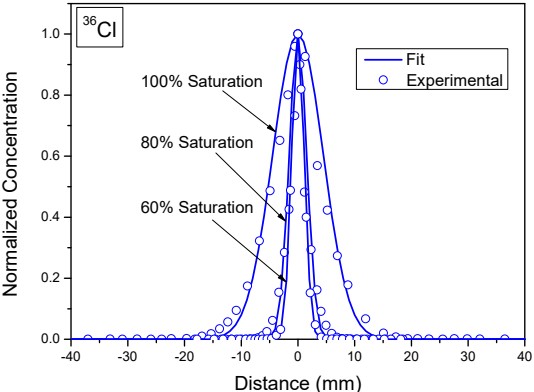

**Figure 6.** Diffusion concentration profiles and fits from IPS tests at three different saturation degrees for HTO and $^{36}$Cl in the sample compacted at 1.65 g/cm$^3$ dry density.

Table 3 shows the summary of the $D_a$ values obtained with the IPS method for HTO and $^{36}$Cl. As the saturation degree decreased from 100% to 60%, $D_a$ values for HTO and $^{36}$Cl decreased by approximately one order of magnitude, with a slightly higher effect in the case of chloride.

**Table 3.** Apparent diffusion coefficient, $D_a$ (m$^2$/s), of HTO and $^{36}$Cl as a function of saturation degree for samples compacted at 1.65 g/cm$^3$ dry density. Errors present one standard deviation.

| Saturation Degree (%) | $D_a$ (m$^2$/s) HTO | $D_a$ (m$^2$/s) $^{36}$Cl |
|---|---|---|
| 100 | $(1.5 \pm 0.1) \times 10^{-9}$ | $(4.6 \pm 0.2) \times 10^{-10}$ |
| 90 | $(4.3 \pm 0.1) \times 10^{-10}$ | $(1.0 \pm 0.1) \times 10^{-10}$ |
| 80 | $(3.7 \pm 0.1) \times 10^{-10}$ | $(5.4 \pm 0.4) \times 10^{-11}$ |
| 70 | $(2.9 \pm 0.1) \times 10^{-10}$ | $(5.2 \pm 0.4) \times 10^{-11}$ |
| 60 | $(2.3 \pm 0.1) \times 10^{-10}$ | $(2.8 \pm 0.1) \times 10^{-11}$ |

For both tracers, the strongest variation in the diffusion coefficient was observed at water saturation degrees from 100% to 90%. Then, the effect was less evident.

On Callovo-Oxfordian clay samples, a similar evolution was observed for HTO where the saturation degree changed from 100% to 81% [14]. Therefore, unsaturated low-porous materials are media in which radionuclide transport is expected to be hindered. Theoretical studies using molecular dynamic analysis for cation diffusion in montmorillonite concluded that diffusion is significantly reduced in unsaturated media compared with fully saturated pore media, which is probably due to changes produced within the water-filled pore network and in pore connectivity [24]. In unsaturated media, the main diffusion paths through the inter-particle space can be discontinuous, and the existence of occluded areas

without groundwater limits mass transport. In addition, water and air coexist in the pore space of unsaturated media.

### 3.3. Instantaneous Planar Source Tests with $^{75}Se(IV)$

IPS tests with $^{75}$Se(IV) were carried out at five different saturation degrees by varying the solid compaction density (1.2, 1.4, and 1.65 g/cm$^3$). Figure 7 shows an example of the experimental concentration profiles obtained for selenium in the sample at a 1.2 g/cm$^3$ dry density and three different saturation degrees.

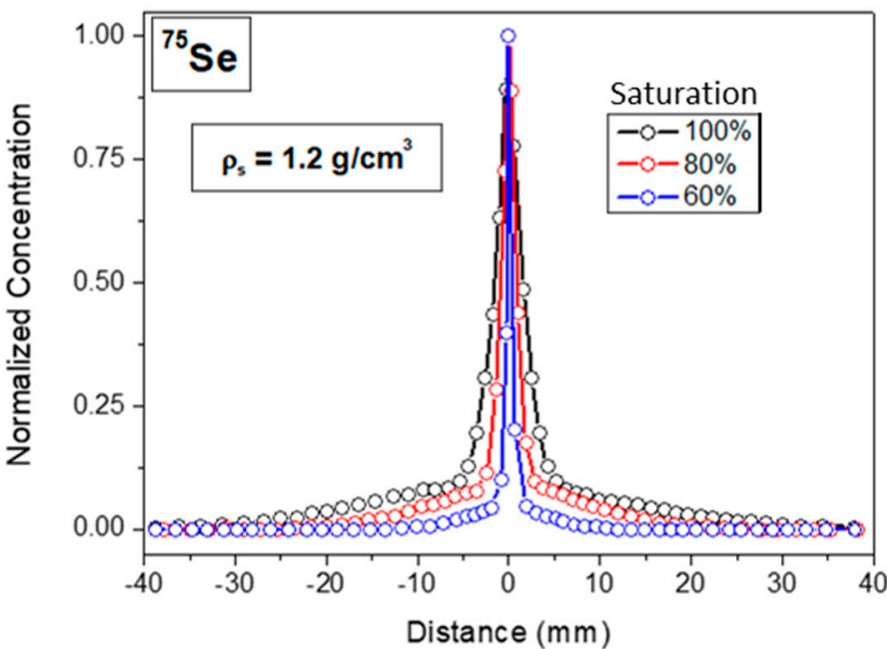

**Figure 7.** Diffusion concentration profiles for the IPS tests at three different saturation degrees for $^{75}$Se at 1.2 g/cm$^3$ dry density.

The shape of the selenium diffusion profiles is significantly different than that observed for the conservative tracers HTO and $^{36}$Cl (Figure 6), which is because selenium diffusion profiles are affected by selenium sorption. Diffusion extends over a wide range within the sample (up to 30 mm), but the central part presents a very narrow peak. This behavior suggests that there must be (at least) two different species controlling selenium diffusion.

Indeed, selenium diffusion profiles cannot be fit with a single diffusion coefficient, such as that obtained with Equation (5). Thus, a hypothesis of the superposition of two profiles to fit selenium data was considered. The equation representing this superposition is given by the following formula:

$$C(x,t) = \frac{M_1}{2 \cdot A \sqrt{\pi \cdot D_{a(high)} \cdot t}} exp\left(-\frac{x^2}{4 \cdot D_{a(high)} \cdot t}\right) + \frac{M_2}{2 \cdot A \sqrt{\pi \cdot D_{a(low)} \cdot t}} exp\left(-\frac{x^2}{4 \cdot D_{a(low)} \cdot t}\right) \qquad (6)$$

Thus, the contribution of two diffusion profiles with "high" and "low" diffusion coefficients ($D_{a(high)}$ and $D_{a(low)}$), represented by Equation (6), was used to simulate the results of selenium. Figure 8 shows the graphical representation of how the two diffusion profiles adequately simulate the $^{75}$Se diffusion tests.

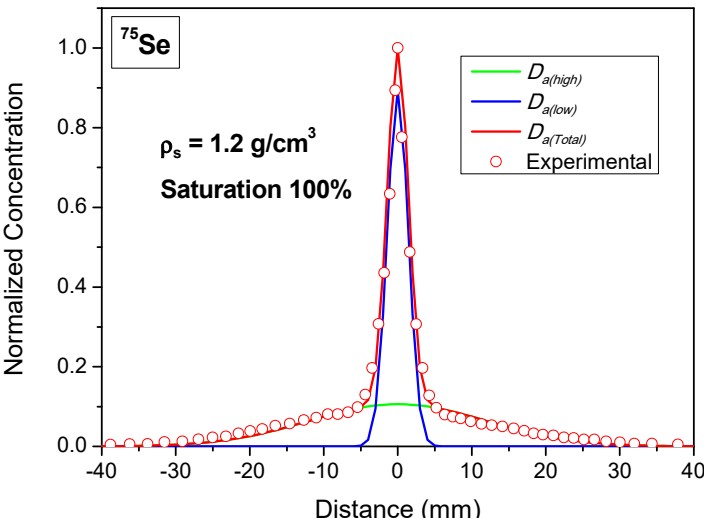

**Figure 8.** Diffusion concentration profile fit using Equation (6) for the superposition of two diffusion profiles.

Table 4 presents the summary of the diffusion coefficients ($D_{a(high)}$ and $D_{a(low)}$) obtained for selenium in the samples at a 1.2, 1.4, and 1.65 g/cm$^3$ dry density and all the considered saturation degrees. All the experimental and modeling data of the sample at 1.2 g/cm$^3$ are provided in the Supplementary Material.

**Table 4.** "High" and "low" $D_a$ (m$^2$/s) of $^{75}$Se as a function of saturation degree (S) and dry density obtained by fitting the experimental results through the application of Equation (6). Errors present one standard deviation.

| S (%) | 1.2 g/cm$^3$ | | 1.4 g/cm$^3$ | | 1.65 g/cm$^3$ | |
|---|---|---|---|---|---|---|
| | **High** | **Low** | **High** | **Low** | **High** | **Low** |
| 100 | $(2.0 \pm 0.4) \times 10^{-10}$ | $(2.9 \pm 0.2) \times 10^{-12}$ | $(1.0 \pm 0.3) \times 10^{-10}$ | $(9.0 \pm 0.1) \times 10^{-13}$ | $(3.6 \pm 1.2) \times 10^{-11}$ | $(2.5 \pm 0.2) \times 10^{-13}$ |
| 90 | $(1.7 \pm 0.2) \times 10^{-10}$ | $(2.5 \pm 0.1) \times 10^{-12}$ | $(7.2 \pm 0.3) \times 10^{-11}$ | $(5.6 \pm 0.1) \times 10^{-13}$ | $(1.5 \pm 0.2) \times 10^{-11}$ | $(1.2 \pm 0.1) \times 10^{-13}$ |
| 80 | $(8.2 \pm 1.1) \times 10^{-11}$ | $(7.8 \pm 0.3) \times 10^{-13}$ | $(4.4 \pm 0.8) \times 10^{-11}$ | $(3.7 \pm 0.2) \times 10^{-13}$ | $(4.3 \pm 1.2) \times 10^{-12}$ | $(4.0 \pm 0.3) \times 10^{-14}$ |
| 70 | $(5.2 \pm 1.4) \times 10^{-11}$ | $(5.1 \pm 0.4) \times 10^{-13}$ | $(1.6 \pm 0.2) \times 10^{-11}$ | $(1.1 \pm 0.1) \times 10^{-13}$ | $(1.9 \pm 0.4) \times 10^{-12}$ | $(1.4 \pm 0.4) \times 10^{-14}$ |
| 60 | $(1.5 \pm 0.4) \times 10^{-11}$ | $(6.9 \pm 0.2) \times 10^{-14}$ | $(5.3 \pm 1.1) \times 10^{-12}$ | $(8.6 \pm 0.5) \times 10^{-14}$ | $(8.3 \pm 0.5) \times 10^{-13}$ | $(8.6 \pm 0.5) \times 10^{-15}$ |

The contribution of the $D_{a(high)}$ fits the wide profile (fast diffusion but less radionuclide concentration), whereas the contribution of $D_{a(low)}$ fits the central sharp peak (slow diffusion but high radionuclide concentration).

The reason why this behavior is observed in the case of selenium is not straightforward to explain. Savoye et al. [25] showed via ion chromatography separation that part of the selenium present in the Eckert & Ziegler commercial $^{75}$Se(IV) radiolabel source was in the form of $^{75}$Se(VI) (5.3% to 6.5%).

A possible redox reaction from Se(IV) to Se(VI) under a neutral-alkaline pH and aerobic conditions has also been reported in different works [26,27]. Se(VI) does not adsorb significantly on natural materials under neutral-alkaline conditions, whereas Se(IV) adsorbs more [28]. The coexistence of two different selenium species with different levels of adsorption—and consequently, different diffusion behaviors—in the solid could give rise to the behavior observed in Figure 7. However, the areas of the corresponding diffusion profiles would indicate a much larger proportion of Se(VI) than that expected by the data of Savoye et al. [25].

The thermodynamic calculations, made with JChess code (EQ3/6 database), indicated that, even if a small degree of oxidation cannot be ruled out, the main selenium species under the conditions of the experiment would be selenite ($SeO_3^{2-}$). Furthermore, considering the mineralogical composition of the rock, the presence of a significant quantity of gypsum

can also be relevant to the overall selenium transport behavior. Lin et al. [29] showed that gypsum has a significant capacity for sequestrating both selenite and selenate in its structure (especially selenite). Incorporation or (co-precipitation) processes of selenium in the gypsum structure may represent an additional retention process for adsorption with different characteristics and kinetics, which could be the reason why two different diffusion profiles are observed.

In general, the difference between $D_{a(high)}$ and $D_{a(low)}$ for Se is approximately two orders of magnitude apart. The diffusion coefficients decrease as the dry density increases, as observed in other clayey rocks [30,31]. Furthermore, as already seen for the conservative tracers, and as is also true in the case of selenium, the diffusion coefficients significantly decreased with the saturation degree. This decrease was more evident at the highest dry density.

## 4. Conclusions

The diffusion of HTO, $^{36}$Cl, and $^{75}$Se(IV) in a sedimentary rock that is under study as a possible host rock for a surface temporal radioactive waste repository was investigated. The through-diffusion technique was used for determining the effective diffusion coefficients for the conservative tracers HTO ($(1.0 \pm 0.3) \times 10^{-10}$ m$^2$/s) and $^{36}$Cl ($(5.7 \pm 0.2) \times 10^{-11}$ m$^2$/s) and for evaluating the anisotropy of the material, which had negligible results.

The instantaneous planar source (IPS) method was used for determining the effects of the water saturation degree on the apparent diffusion coefficients. The decrease in the water saturation degree (from 100% to 60%) implied a significant decrease in the diffusion coefficient for all the radionuclides, hindering their transport.

HTO and $^{36}$Cl concentration profiles at any saturation degree showed a Gaussian shape. They were successfully simulated by applying the diffusion equation for an infinitesimally small thin source where the tracer was homogeneously dispersed. Selenium concentration profiles could be only simulated considering the superposition of two diffusion profiles with diffusion coefficients two orders of magnitude apart. This can be related to different retention mechanisms (surface adsorption and incorporation in the mineral gypsum) or the partial oxidation of selenite to selenate, which also would lead to different retention behaviors.

The IPS method has been demonstrated to be very appropriate for determining low diffusion coefficients in a reasonable time and maintaining moisture conditions during experiments. This methodology can be applied to any other rock where the water saturation degree could be an issue that affects the barrier shield properties meant to protect against contamination spread.

**Supplementary Materials:** The following supporting information can be downloaded at: https://www.mdpi.com/article/10.3390/min13050593/s1. Table S1. Experimental data and fitted results from $^{75}$Se diffusion at 1.2 g/cm$^3$ and 100% saturation; Table S2. Experimental data and fitted results from $^{75}$Se diffusion at 1.2 g/cm$^3$ and 90% saturation; Table S3. Experimental data and fitted results from $^{75}$Se diffusion at 1.2 g/cm$^3$ and 80% saturation; Table S4. Experimental data and fitted results from $^{75}$Se diffusion at 1.2 g/cm$^3$ and 70% saturation; Table S5. Experimental data and fitted results from $^{75}$Se diffusion at 1.2 g/cm$^3$ and 60% saturation.

**Author Contributions:** Conceptualization, M.G.-G. and T.M.; methodology, M.G.-G., M.M. and J.M.; investigation, M.G.-G. and M.M.; writing—original draft preparation, M.G.-G.; writing—review and editing, M.G.-G., U.A. and T.M.; supervision, T.M.; project administration, U.A. and T.M.; funding acquisition, U.A. and T.M. All authors have read and agreed to the published version of the manuscript.

**Funding:** This study was partially supported by the European Union's Horizon 2020 Research and Innovation Program under Grant Agreement no. 847593 (EURAD) and by the Spanish Ministry of Science Innovation MICIN/AEI/10.13039/501100011033 (PID2019–106398GB-I00, ARNO Project).

**Data Availability Statement:** Data can be available upon request.

**Acknowledgments:** Ana María Fernández (CIEMAT) is acknowledged for providing information on the characterization of the rock. We would like to thank the two anonymous reviewers for their work in considerably improving our manuscript.

**Conflicts of Interest:** The authors declare no conflict of interest.

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
