# Peer review of "Analysis of the Role of Water Saturation Degree in HTO, 36Cl, and 75Se Diffusion in Sedimentary Rock"

_minerals, doi:10.3390/min13050593_

Round 1
Reviewer 1 Report
Review on García-Gutiérrez et al: Analysis of the role of water saturationdegree on HTO, 36Cl, and 75Se diffusion in sedimentary rock
General:
At a first glance, the paper looks very good. The topic is interesting and of high relevance. Unfortunately the language is not optimal and the text will need quite some corrections concerning grammar and formulations. I discovered some inconsistencies in the results and/or their interpretation, which need to be corrected or at least well explained before the manuscript can be published.
Major issues:
Chapter 2.2: Was the solution in equilibrium with dolomite? If yes, at which CO2 partial pressure?
Chapter 2.4.1: Why has Se diffusion not been studied via Through-Diffusion? Please explain, or better add experimental results for Se-through diffusion.
Chapter 2.4.2:
Please explain how the filter was applied. Was it wet or dry? If wet: how much water was on it, if dry: how was it dried?
After what period of time were the experiments terminated and the diffusion profiles analyzed?
Figures 4 + 5 + Table 2:
1st y-axis please use standard units Bq/(s*m2), 2nd y-axis is Cumulative activity not mass, and should be in Bq.
How was the flux determined? Usually flux points are allocated in-between cumulative activity and indicate J = dQ/dt. In Fig 5 the linear regression in Fig5 this looks almost perfect, but the flux is not reaching a constant steady state value.
Furthermore, the flux of Cl is higher than that of HTO. In Table 2, De(Cl) is smaller than De(HTO), which is expected, but does not fit with the data shown in Figs 4 + 5.
Alphas should be reported in Table 2. Also here data indicate an unexpected trend, that alpha(Cl) looks larger than alpha(HTO).
Table 2 should also report an anisotropy factor De(parallel)/De(perpendicular)
Chapter 3.2 + Table 3:
Values reported in Table 3 seem consistent in itself, but how do they fit with the through diffusion results? IPS results for HTO at 100% saturation show Da = 1.5e-9, which is about 15 times higher than De(TDS). Da = De/alpha, thus, alpha should be about 0.07, which is small, but might be plausible for a highly compacted rock.
For Cl, IPS results in Da = 4.6e-10. With Da=De/alpha and De(TD) = 6e-11 we get alpha = 0.13, which is higher than the value for HTO.
So please either fix this or explain thoroughly where these discrepancies come from and what they mean. How representative are the IPS measurements on the crushed and re-compacted rock? How can they be related to the through-diffusion results?
Chapter 3.3, Figs 7 + 8:
Please show all Se data, maybe in a supporting information file.
In Fig 8 the x-axis units are wrong, should be m, but better use consistently mm as in other Figures.
In my opinion these major issues should be fixed first, afterwards a second round of reviews may decide whether the manuscript can be published.
Reviewer 2 Report
see attached file

Round 2
Reviewer 1 Report
The authors largely addressed the reviewer suggestions in an apropriate way. The following issues remain:
In my opinion the authors must show all Se-diffusion data, at least in a supporting information file. The authors should report the M1/M2 and M1+M2 values for the fits to the Se diffusion data, so the reader can evaluate the extent of the fast versus slow diffusion and assess whether this fits with the explanation given in the text.
In table 2 the authors should report the uncertainty of the anisotropy factor, and consider corresponding significant digits.
Once these issues are fixed the paper is ready for publication after a minor spell check.
Author Response
Comments and Suggestions for Authors
The authors largely addressed the reviewer suggestions in an apropriate way. The following issues remain:
In my opinion the authors must show all Se-diffusion data, at least in a supporting information file. The authors should report the M1/M2 and M1+M2 values for the fits to the Se diffusion data, so the reader can evaluate the extent of the fast versus slow diffusion and assess whether this fits with the explanation given in the text.
Ok. Done. Data have been included as Supplementary material
In table 2 the authors should report the uncertainty of the anisotropy factor, and consider corresponding significant digits.
Ok. Done.
Once these issues are fixed the paper is ready for publication after a minor spell check.
Thanks for you efforts.

Author Response
Manuscript ID: minerals-2215373-revised
Title: “Analysis of the role of water saturation degree on HTO, 36Cl, and 75Se diffusion in sedimentary rock”
Authors: Miguel García-Gutiérrez, Manuel Mingarro, Jesús Morejón, Ursula Alonso, Tiziana Missana
The authors clearly improved the manuscript and addressed almost all of my comments by adding missing information. However, in some cases, I have the impression I misled the authors with my comments and the added statements went to an improvement for the worse. Furthermore, the requested inclusion of uncertainties in the figures is still not in the manuscript. I tried to clarify these points in this review.
Comments:
Abstract:
Line 13: Just for readability “though-diffusion (TD) method and … (IPS) method.”
Line 15: comma after “De”
Line 25: “satisfactorily fitted”
DONE
Introduction:
Line 31: I think, my comment was misleading you. “disposal” was correct, but “scenario” not.
“scenario” does neither go with “disposal” nor with “repository”. Surface radioactive waste disposal is a concept of waste management, where one scenario would be the radionuclide migration into the surrounding soils.
CORRECTED
Line 37: comma after “[2]”
Line 38: replace “During” with “For”
DONE
Lines 39-40: Also in this case, you misunderstood me: “ventilation of drifts and shafts will be active, and this will desiccate the gallery walls consisting of shot concrete and clay host rock.”; “For instance” not “Furthermore”
CORRECTED
Line 63: comma before “too”
Line 75: hyphen “non-sorbing”
Line 77: comma after “anion” and correct term is “anion exclusion”
- DONE
Line 78: word order “selenium, which is in form of ??? a slightly sorbing tracer”; Se(IV) is the oxidation state, its sorption capability depends on its actual chemical form in the water. Is it under the investigated conditions, an anion, a cation, or a neutral species?
Yes, it was indicated that selenite is an oxyanion
Line 126: “was” not “were”
Line 156: “;” after “corner”
Line 164: delete period after the colon
Line 166: comma after “tracers”
Line 173: comma after “90%”
Line 175: “a” not “as”
Line 176: you gave the dry density information twice in one sentence, delete “were carried out with the three different dry densities and”
Line 178: comma after “7” and comma after “1.4”
Line 189: Please delete the time and density information. You provided this already on page 5.
Line 200: comma after “source”
DONE
Equations (3) and (4): Not everything has to be presented in italic, only the symbols, not the parentheses, numbers, and signs. Look at equation (5). There, it is correct.
Corrected
Line 217: “as” not “has”
OK
Results and discussion:
Figures 4 and 5: I see that you present the data in terms of Bq in the revised version, which is better for comparison with literature data. However, I am still missing the uncertainties of the experimental points and the fit of the accumulated mass. The experimental uncertainty is composed of the systematic errors of your applied measurement methods (LSC, gamma counting, distance measurements, time measurements), which is provided by the instruments or should be estimated by you, and the statistical error of your duplicate experiments (standard deviation, confidence interval). In that way, you obtain error bars for your experimental data. Concerning the accumulated mass plot, you can calculate the uncertainty of De and α with the error propagation law considering the uncertainty of L, A, C0, t, and Q. This is all doable with a simple Excel calculation. Plots considering these uncertainties should be given in the figures.
The uncertainties have been included and the Figure have been modified.
Line 230: add “In addition, Table 2…”
Lines 236-240: This is a contradicting statement. Because you see systematic differences, there is negligible anisotropy of the material when considering the uncertainties. Please, rewrite this part.
Rewritten
Line 242: I am glad, that the porosities are provided in the revised version. It would be even better, if these data had been discussed in comparison with literature data and in connection with the provided porosity of 29.8 % of the raw material. Why is there such a difference?
The data of the raw material was taken from a previous characterization report for all the samples of the site. The measurements reported in this report were carried out by mercury intrusion porosimetry. In the report was also indicated that the porosity measured by MIB vary between 3 and 35 % depending on the gypsum and lime distribution, showing high heterogeneity degree.
Lines 279-280: wording, either you write “varying the solid compaction density” (singular) or you write “various solid compaction densities” (plural), the form in the current version is not correct
Equation (6): right parenthesis is missing after “(low”
Line 302: instead of eq. (3), should not be that eq. (6)?
Line 327: What would be, based on thermodynamic calculations, the dominant Se(VI) species, when Se(IV), present as anion, is sorbing?
Line 335: “decreased” not “decreases”
DONE
Conclusions:
Line 343: What is the conclusion concerning anisotropy and the through-diffusion experiments in general?
Information added
Line 347: delete space between “100” and “%”; “water saturation degree” singular
Line 350: new sentence “shape. They were …”
DONE

Round 3
Author Response
Manuscript ID: minerals-2215373-revised
Title: “Analysis of the role of water saturation degree on HTO, 36Cl, and 75Se diffusion in sedimentary rock”
Authors: Miguel García-Gutiérrez, Manuel Mingarro, Jesús Morejón, Ursula Alonso, Tiziana Missana
The authors did a very good job considering my comments of two reviews. In this review, I just have mainly minor edits. But I still have a problem with the data presentation in the tables. I do not understand how the uncertainties were calculated. Here, I would need a clarifying statement from the authors. See my comments below.
Yes, the referee is right. The errors of the diffusion coefficients were calculated by the arithmetic mean of the experimental values obtained, whereas the error propagation law was used to calculate the uncertainty of the anisotropy factor.
Comments:
Abstract:
Line 17: “This revealed a small anisotropy of the rock.” This is contradicting to your conclusions, where you state, that the anisotropy is negligible.
OK. We changed “small” by “negligible”.
Line 23: Delete the second “by”, it should be “…decreased by at least one …”
OK. Done.
Introduction:
Lines 39-40: The sentence is still not understandable. Maybe, this is what you intended to say: “…, the ventilation of drifts and shafts will be active, which will desiccate the walls of clay host rock and shot concrete [3].”
OK. Rewritten.
Line 77: comma after “tracer”
OK. Done.
Materials and Methods:
Line 103: “performed” is mentioned twice in one line; please delete the added phrase “performed in the total rock”, since this text is connected to the previous paragraph, it is clear at this point that the bore core material is meant.
OK.Done.
Line 126: Since you already clearly stated in Line 97, that all experiments were conducted under aerobic conditions, it is not necessary to mention that the water was prepared under oxic conditions at this point again. I would suggest deleting the sentence here.
OK. Deleted.
Results and discussion:
Line 234: large parenthesis after “(perpendicular)” is missing
OK. Corrected.
Table 2: Since you state here, that the uncertainties are calculated by the error propagation law and since you have run all experiments in duplicate, the values and uncertainties of which cells are given here? Have you not calculated an arithmetic mean, too? If yes, you should present that instead or better in combination with the error propagation law. Since the uncertainty values have not changed to the previous manuscript versions, the error propagation law was probably just used to calculate the uncertainty of the anisotropy factor. The other uncertainty values seem to be a result of arithmetic mean calculations of results from two experiments. In summary, it is not clear to me how the uncertainties were obtained, please clarify. In addition, the naming of the column headers should be adjusted. Not every value in the columns is a De. I suggest moving “De (m2/s)” of the second and third column to the first two rows in the left column, leaving just “HTO” and “36Cl” in the table header and naming the first two rows “De (m2/s) parallel” and “De (m2/s) perpendicular”.
Yes, the referee is right, as we commented above. A sentence is included in the manuscript to clarify that, and the comment in the caption has been deleted.
OK. We modified the headers in the column.
Lines 239-243: There is still the same problem as in the previous manuscript version. First, you mention “systematic differences” which means you have clearly a bias in the system, which causes a real reproducible difference depending on bedding direction. Then, you state the differences are negligible. These are contradicting statements. Never discuss and compare values without their associated uncertainty. Therefore, please delete the first three lines and just state that an anisotropic diffusion behavior was not observed for HTO. However, under consideration of the uncertainties, a slight anisotropy seems to be present for 36Cl diffusion, which is in the current version not discussed. You write in the abstract from a small anisotropy effect, here and in the conclusions from a negligible effect. The statements on these three positions should be all consistent.
We think that the anisotropy is negligible and we have written it in the text.
Line 245: Still the discussion of the porosity determined by diffusion compared to the porosity determined by mercury intrusion porosimetry is missing. You provide a well explaining statement in your response to me. Please provide this information also in the manuscript.
OK. Done.
Line 250: “In contrast to the results for the sedimentary rock investigated in the present study, they found anisotropy …” You should highlight this important result of your study more (absence of anisotropic behavior in the sedimentary rock from the site in Spain).
OK. Done.
Table 3: “Uncertainty calculated by error propagation law.” See statement Table 2. It is not clear to me, how the uncertainties were calculated. You state that they are obtained using the error propagation law (considering just systematic errors), which I suggested to use for calculation of uncertainties in my last review. But in the current version, it seems you just added this statement to the table and you did not do the actual calculation, since the uncertainties have not changed to the previous version. You have run duplicate experiments. Therefore, I assumed that the uncertainties in the table are the random errors you obtain by running multiple experiments. Please can you clarify how you considered the results of duplicate experiments (standard deviation, confidence limit) as well as the systematic error, where you used the error propagation law for, in your uncertainty calculation and how this leads to unchanged uncertainties?
Yes, as stated before, the errors presented in this table represent the experimental standard deviation. A sentence is included in the manuscript to clarify that.
Table 4: Are the uncertainties here the standard deviation or confidence limits obtained after two measurements? Or are they calculated with the error propagation law?
Same as Table 3.
Conclusions:
Lines 354-355: “HTO and 36Cl concentration profiles at any saturation degree showed a Gaussian shape.”
OK. Corrected.
Round 4
Author Response
Dear reviewer, we have inserted the last correction in the marked versión. We do not understand the last comment (Line 365, delete the second and) because one "and" was already deleted. Thanks a lot for your huge effort in improving our manuscript.